# A Survey of Patients’ Opinions and Preferences on the Use of E-Prescriptions in Poland

**DOI:** 10.3390/ijerph18189769

**Published:** 2021-09-16

**Authors:** Natalia Wrzosek, Agnieszka Zimmermann, Łukasz Balwicki

**Affiliations:** 1Department of Medical and Pharmacy Law, Medical University of Gdańsk, 80-210 Gdańsk, Poland; natalia.wrzosek@gumed.edu.pl; 2Department of Public Health & Social Medicine, Medical University of Gdańsk, 80-210 Gdańsk, Poland; balwicki@gumed.edu.pl

**Keywords:** e-prescription, e-health, pharmacy

## Abstract

E-prescription is already used in many countries, improving the standard of patient care. Officially, from 8 January 2020 e-prescribing has been obligated in Poland. Physicians’ and pharmacists’ opinions on e-prescribing have been widely researched and reported in the literature. In contrast, patients’ perception has, to date, received little attention. For this reason, the aim of this study was to find the features and functionalities of e-prescribing that are desired by the public and influence the positive evaluation of this tool, according to patient opinion. In order to obtain data, a questionnaire was completed by 456 randomly selected adults. The obtained results indicated that only eight people (1.8%) did not know what e-prescription is. Of the remaining 448 individuals, 72.1% prefer e-prescription because it is more convenient for them. Most patients (62.1%) also recognize that e-prescribing makes it easier to purchase medications on behalf of another patient. Based on the study, it can be concluded that e-prescription is well evaluated by Polish patients. A large percentage of respondents were positive about obtaining prescriptions for continued treatment, without a personal doctor visit. Therefore, it is reasonable to maintain the possibility of such contact with a physician. The most popular, and preferred, method of receiving e-prescriptions is via SMS. However, it is necessary to offer different options for obtaining prescriptions, to meet the needs of different populations.

## 1. Introduction

The digitalization of healthcare is a common direction of development in many countries around the world, aiming at improving the standard of patient care and increasing their safety during the treatment process. Electronic prescription, as a basic tool of e-health, is already used in Canada, USA, United Kingdom, Australia, Spain, Japan, Sweden, and Denmark [1]. An e-health service is being introduced in all EU countries. In addition, by 2025, cross-border health services will be gradually implemented in 25 EU countries: Austria, Belgium [2], Croatia, Cyprus, Czech Republic, Estonia, Finland, France, Germany, Greece, Hungary, Ireland, Italy, Lithuania, Luxembourg, Malta, the Netherlands, Poland, Portugal, Slovenia, Spain, Sweden, Slovakia, Latvia, and Bulgaria. European Union citizens will be able to obtain their medication in a pharmacy located in another EU country, thanks to the online transfer of their electronic prescription from their country of residence where they are affiliated, to their country of travel [3]. The electronic form of prescription is also gaining popularity in countries such as India [4] and Nigeria [5].

Electronic prescriptions (hereinafter referred to as e-prescriptions) are one of the key e-services in the ongoing process of Polish healthcare system informatization. E-prescriptions are associated with a number of benefits for both patients and medical staff. Introducing e-prescriptions into the Polish healthcare system was supposed to eliminate the problem of illegible prescriptions and, as a consequence, have a positive impact on patient treatment. This is because patients will not be forced to see a doctor again to correct an illegible document. In addition, patient safety is expected to increase, by reducing the possibility of dispensing the wrong medication. The changes are designed to improve care for chronically ill patients who do not have to make an in-person doctor’s appointment to receive a prescription for continued treatment. The e-prescription can be issued remotely via an online consultation. This is estimated to result in significant time savings for both patients and physicians [6]. Moreover, an electronic prescription allows a prescription to be filled more easily for another person. This functionality proved especially valuable during the COVID-19 pandemic, when many people were in isolation or quarantine and were unable to go to the pharmacy in person. Among the other patient benefits of e-prescriptions is the ability to purchase each prescribed medicine in a different pharmacy. With this facility, it will be easier for patients to obtain all the necessary medicinal products independently, and the pharmacy’s stock should not affect the continuity of the patient’s treatment. The patient also has the option of partially purchasing drugs, but it is important that subsequent packages of the same drug must be continued at the same pharmacy.

An additional advantage of developing the idea of e-health is the introduction of the online patient account (OPA). This program allows the collection of patient health data in one place, including how often prescriptions were issued. The patient, medical staff, and persons authorized by the patient can get access to the history of prescribed medicines, which is expected to increase the level of quality and safety of treatment [7].

The e-prescription system in Poland has been gradually implemented since 2018. Officially, from 8 January 2020 e-prescribing has been obligated in Poland. There are a small number situations in which it is permissible to issue a prescription in paper form. Electronic prescriptions issued during a medical visit are saved on an online platform called the P1 system. All pharmacies, clinics, and medical offices in Poland are obligatorily connected to the P1 system. The patient can choose any pharmacy to fill his prescription. To download a prescription from the P1 system, access data should be provided to the pharmacist, which the patient receives from the doctor in several possible ways. According to Polish pharmaceutical law, a paper-based prescription can be issued if the physician does not have access to the nationwide online system for confirming prescriptions (P1 system), or for a person of unknown identity [8].

A Polish patient can receive an e-prescription during a medical visit in several ways. One of these is an informational printout that the doctor can hand out on paper or send as an electronic file to the patient’s email address. This document contains a barcode which is used to retrieve the prescription from the nationwide P1 system by the pharmacist in the pharmacy. Another form of access is the four-digit access code, which in combination with the patient’s PESEL number also allows the prescription to be dispensed at a pharmacy. The PESEL number is an eleven-digit numeric symbol that uniquely identifies a specific person. The number includes the date of birth, serial number, gender, and a check number. It is given by the minister responsible for computerization. The access code may be given to the patient via SMS or in another agreed form. Along with the code, information about the prescribed drug must be provided, along with instructions on the need to provide a PESEL number at the pharmacy. [9]. This is very important because, in the first period of operation of e-prescription, a common problem was the lack of awareness about the need to show ID at the pharmacy when buying drugs, so patients, especially the elderly, were confused and impeded in buying drugs at the pharmacy [10]. Having only a four-digit code without entering the patient’s PESEL number is not sufficient to download a prescription from the P1 system. A free government mobile application, mCitizen, is also offered to Polish patients to access all e-prescriptions issued [11].

In the era of widespread access to the Internet and new technologies, patients are becoming more involved in the process of their treatment and are looking for modern methods of contact with medical staff. The development of technology and widespread access to the Internet has enabled new methods of communication between doctor and patient [12,13]. However, in countries where e-prescriptions have been in place for a long time, practical problems, failure to implement, and the creation of additional barriers for patients are observed [14,15]. Physicians’ and pharmacists’ opinions on the advantages and disadvantages of e-prescribing have been widely researched and reported in the literature [8,16]. In contrast, patients’ perception of e-prescribing is, as yet, an issue that has received little attention [17,18]. This is very important, because satisfied patients are more likely to continue using health care services, as well as more likely to maintain contact with their doctor, adhere to treatment, and actively participate in their own treatment process [12].

For this reason, this study analyzes the opinions of the recipients of the e-prescription service, i.e., patients. It constitutes another stage of a research project evaluating the process of implementation of e-prescription in the Polish health care system. In phase one of the project, a preliminary analysis was made of the opinions and experiences of medical personnel, who work with the e-prescription system on a daily basis [12]. In this study, which is a continuation of the research project, the main goal was to evaluate the opinions and the degree of acceptance of the technology by all the entities participating in the issuance and dispensation of e-prescription, as well as the opinions of the recipients of the service, the patients, in order to comprehensively describe the process of implementation of e-prescription and analyze its usefulness. Considering the dynamic development of e-health in Poland and plans to implement further instruments related to digitization of healthcare, determining patients’ opinions on this topic seems to be a pressing matter.

### Aim of the Study

The overarching goal of this study was to find the features and functionalities of e-prescribing that are desired by the public and influence the positive evaluation of this tool. We also searched for disadvantages of e-prescription and difficulties associated with it. Additionally, this study aimed to record the opinions and to estimate the level of satisfaction of Polish patients regarding the use of e-prescription.

## 2. Materials and Methods

### 2.1. Research Tools

In order to obtain data for the necessary analyses, a proprietary questionnaire was created and distributed, both electronically and in paper form. The questionnaire included questions measuring patient satisfaction with the use of e-prescribing and knowledge and opinions about the tool. The survey consisted of 17 questions and a metric verifying the respondent’s age, gender, education level, and place of residence. (Appendix A). A five-point Likert scale was used in the opinion and satisfaction questions. On the other hand, short answers were required in questions regarding the experience of using prescription, as well as using pharmacy services. Two filter questions were used, which optionally exempted people who had no experience of using an e-prescription or Internet Patient Account from answering some parts of the survey. The tool was validated by five independent experts from the Medical University of Gdansk with experience in social research. Content validity ratio (CVR) was determined for each question separately. Questionnaire items for which the CVR was less than 0.9 were removed or corrected according to the experts’ suggestions.

### 2.2. Study Setting

The goal was to obtain the largest possible group of respondents; therefore, the form was made available among various social groups throughout Poland. Many communication channels were used, including local government websites, patient associations, and online health forums. Responses were collected between May 2020 and January 2021, and then statistically analyzed.

The research project was approved by the Independent Bioethics Committee for Scientific Research operating at the Medical University of Gdansk (opinion number NKBBN/664/2018). 

### 2.3. Statistical Analyses

Data analyses were performed using IBM SPSS software (IBM, Armonk, NY, USA). To evaluate the presence of correlations between variables and to assess the significance of differences in frequency distributions between groups, contingency tables were utilized and chi-squared tests were applied. The limit of statistical significance was established at *p* < 0.05.

## 3. Results

### 3.1. Characteristics of Study Group

A total of 456 people participated in the study, of which the majority (345, 75.7%) were women. The highest percentage of respondents (41.9%) was represented by people under 30 years of age, followed by 32.5% of people in the age range of 31–50 years. People aged 51–70 and over 70 were the least numerous and constituted 19.7% and 5.9%, respectively, of the total study group. Most of the respondents had higher (67.8%) or secondary (27.6%) education. People representing primary or vocational education constituted 4.6% of all respondents (Table 1). Response rates have not been calculated, as the questionnaire was distributed through many channels, both online and in paper form, in order to obtain a high number of completed forms.

The study group was almost equally represented by healthy people (52%) and people suffering from chronic diseases (48%), which was verified by the answers to the question about the method of taking medicines. While, 47.4% of patients declared that they take medications mainly on an ad hoc basis, the others (52.6%) reported constant use of medications. 

### 3.2. Knowledge and Use of E-Prescription

Most of the patients filled a prescription at the pharmacy from one to four times during six months. Statistically significant relationships were found between age and the number of times one dispensed a prescription for someone else (*p* = 0.001), which indicated that prescriptions for another person were least frequently filled by those aged over 70 years and most frequently by those aged 31–50 years. In addition, it was shown that residents of the largest cities were least likely to fill a prescription for another person and rural residents were most likely to do so. This relationship was of statistical significance (*p* = 0.007) (Table 2).

Nearly 40% (39.5%) of respondents had had their own prescription filled by someone else in the past six months. The opposite was true for the 58.8% who happened to fill a prescription for another person. The most common prescriptions were for parents (158 cases), or partners or spouses (169 cases) (Table 3).

When asked if the patient knew what an e-prescription was, only eight people (1.8%) gave a negative answer. This was a filtering question. Of the remaining 448 individuals, 89.5% (401) reported filling a minimum of one e-prescription in the past six months. Statistically significant correlations were found between the variable defining gender and declaration of knowledge of the electronic prescription (*p* = 0.011). Women were more likely than men to declare knowledge of the e-prescription concept (Pearson’s chi-squared coefficient: 6.438; df = 1, odds ratio = 5.38, confidence level: 1.26, 22.87)

### 3.3. Prescription Preferences

Those who are aware of what e-prescribing is would overwhelmingly prefer such prescriptions if given the choice (323 respondents, 72.1%). For 108 respondents (22.3%), the form of drug prescription does not matter, and only 25 (5.6%) supporters of traditional paper-based prescriptions were reported. 

Respondents were given the opportunity to justify their preference for the form of prescription in writing. The answer to this question was open and voluntary. The participants could give various arguments. The obtained answers have been summed and grouped according to recurring arguments. A total of 431 arguments for electronic prescriptions and 35 arguments from supporters of paper prescriptions were obtained. Of the responses indicating an electronic version, the most common justification was the great convenience of e-prescribing (124 responses, 28.8% of all arguments). No risk of losing the prescription and no need for a doctor’s visit were also frequently cited advantages of e-prescriptions (70 and 43 responses, respectively). Those who would prefer to receive paper-based prescriptions most often justified this with easier access to the list of prescribed medications (13 responses, 35%). Other arguments in favor of traditional prescribing were having more control over prescribed medications (6 responses, 16%) and that e-prescribing proves to be a technological challenge for the elderly (5 responses, 14%). It was not mandatory to answer this question (Table 4).

Respondents were asked to respond to the statements given in the survey describing the features of e-prescriptions. The results indicate that patients are most strongly in favor of the statement that e-prescribing has a positive impact on ecology, where 67.1% of respondents strongly agreed with this statement (86.6% positive responses in total). Most patients (62.1%) also recognize that e-prescribing makes it easier to purchase medications on behalf of another patient (19.1%, rather yes, 43%, definitely yes). The vast majority of respondents (61.6%) were not afraid of having their privacy violated when filling an e-prescription, but 20.6% of respondents had such concerns. For both the question on whether e-prescribing allows for greater control over prescribed medications and the question on the impact of e-prescribing on access to medications a similar distribution of responses was observed, with a preponderance of supportive responses. Most people (36%) could not clearly assess whether they agreed with the view that e-prescribing increases patient autonomy in the treatment process. However, a statistically significant relationship was observed indicating that those using OPA were more likely to select the response “rather agree” or “strongly agree” (Pearson’s Chi-squared coefficient: 13.53; df = 4; *p* = 0.009). (Table 5).

On a five-point scale where 1 means “very bad” and 5 means “very good”, more than half of patients (249, 54.5%) rated their satisfaction with using e-prescribing by giving the highest rating. Only one person rated their level of satisfaction as the lowest. The mean satisfaction rating was 4.37 (Median: 5, standard deviation: 0.8) (Figure 1)

SMS is the most common way to communicate the prescription to the patient, as reported by 254 respondents (55.7%). The second most popular form is a printout with a barcode (19.1%). The traditional paper-based prescription was in the third place, as indicated by 40 (8.8%) participants. Less common but current practices were email (23 people, 5%), prescription code given verbally during an online consultation (19 people, 4.2%) or in-person visit (8 people, 1.8%), and a paper with the prescription code (7 people, 1.5%). Six people reported using the mCitizen app to gain access to a prescription (1.3%) or the OPA (5 people, 1.1%). There were seven unclassifiable responses (1.54%) (Table 6).

For patients, the most convenient form of receiving a prescription was via a text message sent to a phone number (SMS) (69.3%), with a barcode information printout being largely accepted (14%). Email or paper prescription were the least preferred forms of prescription delivery, as perceived by the patients (9.9% and 6.8%, respectively).

## 4. Discussion

This study is the first to address the topic of e-prescription as perceived by patients in Poland. A large study group allows drawing conclusions on this topic. The topic discussed in the study is current and touches upon the current issues of digitization of health care.

The results of this study indicate that the ability to fill prescriptions on behalf of another person is popular among respondents. More than 60% felt that e-prescribing made this task easier. The ability to authorize a family member to access the data contained in the OPA is an additional improvement in this regard. The authorized person has access to all prescriptions issued for the patient, so there is no need to provide the prescription code every time, which could often lead to confusion and problems with obtaining medicines [19]. Prescriptions for another person were least likely to be filled by those aged 70 or older and most likely to be filled by those aged 31–50. This correlation may indicate the need for younger people to support the elderly, who may be experiencing e-exclusion; also known as a social exclusion in the information society [20]. This phenomenon can be divided into two categories: based on limited access to hardware and software and the ability to use them; or based on psychological reasons, such as low self-esteem, fears, resistance, privacy concerns, and level of skill in using hardware, applications, or the Internet. According to research conducted by the Center for Social Opinion Research, the most important factor affecting digital exclusion is age, and this includes the majority of those surveyed over the age of 54. Due to the ageing population, it seems necessary to adapt digital solutions to this age group. [20]

This study showed that people living in rural areas are more likely to fill prescriptions on behalf of another person than residents of larger cities. This may suggest limited access to well-stocked pharmacies in rural areas and reduced mobility for many patients living there. Residents who own cars in rural communities often help others take care of their needs by filling prescriptions at a pharmacy located in the nearest town. The number of pharmacies located in rural communities has steadily declined since 2017. In two years, 215 such facilities have closed, which is also one of the factors affecting health inequalities among rural and city residents [21,22].

Survey respondents overwhelmingly reported that they know what an e-prescription is. In addition, the vast majority of respondents reported dispensing a minimum of one such prescription in the past six months. It is worth noting that a subjective assessment of one’s familiarity with e-prescribing does not necessarily reflect the actual state of knowledge. In a similar study conducted in Belgium, the majority (68, 81%) of people surveyed declared that they knew what e-prescription is [23]. However, 34% of respondents declared that their physician does not use e-prescribing. Such answers were given mainly by patients over 50 years of age. The authors explain that as long as patients receive a paper information printout at a doctor’s visit, they do not realize that this is a type of e-prescription [23]. Similar conclusions can be drawn from a 2012 study in Indiana, where structured interviews were conducted with patients visiting a community pharmacy. Most of the interviewees could not explain on their own what e-prescription was, or the answers declared were wrong. Most commonly, e-prescribing was equated with using a computer and printer instead of paper and pen when prescribing therapy. Some have associated e-prescriptions with the ability to automatically dispense medications in some pharmacies [24].

Patients who declared that they know what e-prescription is evaluated it positively. The average satisfaction rating on a scale of 1 to 5 was 4.37. In addition, respondents overwhelmingly (72.1%) would prefer to have medications prescribed in this manner. The survey noted only 5.6% supporters of the traditional paper prescription. A higher percentage of proponents of e-prescribing was reported in a study conducted in the United States in the state of Pennsylvania, where as many as 80% of participants preferred using e-prescriptions over paper prescriptions. There, too, over 92% of patients said they were highly satisfied with the use of e-prescribing [17]. The prevalence of proponents of e-prescribing has also been reported in other states [25]. E-prescribing is also better accepted among Swedish patients, where only 4.2% of respondents would like to replace e-prescriptions with traditional paper versions. In that country, it was noted that among the age group of 25–39 years a higher percentage of respondents declared positive experiences with e-prescription than among those above 75 years of age [12]. No such correlation was noted in Poland. However, given the right to choose, 51% of Belgian patients would prefer to receive a printed version of the prescription. This answer was given mainly by patients with chronic illnesses and those using more drugs. In that country only 37% of the surveyed patients had a positive opinion about e-prescription. It is worth emphasizing that the method of providing access to the prescription of the patient and its dispensing in a pharmacy using an information printout is very similar in Poland and in Belgium [2,23].

Patients who stated that they prefer e-prescriptions to paper-based prescriptions most often argued that they are more convenient to use. E-prescriptions, according to patients, allow less frequent visits to the doctor and are more difficult to lose or destroy. The same features were most often mentioned by patients in Pennsylvania as advantages of e-prescription [17]. Belgians most often presented the impeccable legibility of the e-prescription as its greatest asset. Among other advantages of e-prescribing, less environmental pollution, as well as lower risk of prescription forgery, were also mentioned [23]. The vast majority of respondents in this study also agreed that e-prescription has a positive impact on ecology. In addition, the vast majority of Poles believe that the use of e-prescription facilitates access to medicines. The same opinion was held by 85% of Swedish patients answering a similar question in their country [12].

Those who prefer paper-based prescriptions most often justified it by the easier access to the list of prescribed drugs. In addition, it was argued that the traditional form of prescription allows better control over the drugs prescribed and e-prescription is a technological challenge for the elderly. Lack of control over the validity of the prescription or the form of the prescribed drugs was also the main argument sustaining the negative opinion of e-prescription in studies conducted in both Belgium and Pennsylvania and Indiana (USA) [17,23,24]. It is worth mentioning that e-prescribing in Pennsylvania and Indiana involves direct mailing of the patient’s prescription to a pharmacy of the patient’s choice. The patient only receives information about when the medications are ready for pick-up. It appears that 5.3% of respondents were unsure if the medications they receive from the pharmacy are definitely the ones the doctor prescribed electronically. However, this is a small percentage compared to the 93% who expressed high satisfaction with this solution [8]. For many patients, the paper prescription serves as a reminder to buy their medications [23]. The same argument was also made by patients surveyed in a study conducted in the state of Rhode Island. Respondents preferred paper prescriptions because they help them remember to visit the pharmacy. Through years of experience with paper prescriptions, the vast majority of older patients still expect to receive written information from their physician, whether or not an e-prescription has been issued [25].

In the present study, the vast majority of respondents were not afraid of having their privacy violated during e-prescribing, but 20.6% of the respondents had such concerns. Similar results were obtained in a study conducted in Belgium, where a significant proportion of patients surveyed expressed doubts about securing their privacy when using e-prescriptions [23]. Meanwhile, an article was published in 2021 that analyzed and compared e-prescribing systems in eight different countries, including six European countries. Consideration was given to ensuring digital security and following protocols related to user privacy. According to the authors, the guarantee of secure patient data processing adheres to Health Level Seven International (HL7) procedures, which describe the standard for digital information exchange in medical environments [1]. The Center for Health Information Systems, which is the administrator of the Medical Information System collecting medical data in Poland, has implemented the HL7 standards, in accordance with the detailed instructions [7,26]. Doubts about ensuring data privacy are one of the main barriers hindering the implementation of e-health tools both in Poland and other countries. It is worth emphasizing, however, that European countries, unlike the United States, are obliged to observe very strict regulations concerning personal data protection [27]. 

More than one-third of the survey participants could not clearly state whether e-prescribing positively affects their autonomy in the treatment process. In contrast, it was noted that active OPA users were more likely to provide affirmative responses. In a similar study conducted in Belgium, where the e-prescription system is very similar to the Polish one, patients were mostly positive about the impact of e-prescribing on patient autonomy. However, literature reports indicate that patients receiving e-prescriptions are more likely to check their prescribed medications, which may have a positive impact on the level of awareness of the treatment process. In addition, it has been proven that these patients are more likely to talk to their doctor about their medications than those receiving paper prescriptions [25]. Given the fact that there is still a portion of paper prescription adherents for whom the traditional version of prescribing helps to control their treatment process and pharmacotherapy, special attention should be paid to the role of the pharmacist and his/her informational and educational role when dispensing medicinal products. With widespread online consultations, the pharmacist is often the only health care professional with whom the patient has the opportunity to meet in person. It is important for the pharmacy staff to ensure that the patient is aware of their treatment and therapy. This is also another argument for the need to quickly implement pharmaceutical care in pharmacies.

The most popular, as well as the most desired, form of delivering a prescription to a patient in Poland is a text message sent to a phone number. This form is accepted by the majority of respondents, regardless of their level of education. According to a report commissioned by the Office of Electronic Communications in Poland, nine out of ten adult Poles use a cell phone. Nearly three-quarters of them own a smartphone, through which 62% of users access the mobile Internet [28]. In second place in terms of popularity and acceptability was the barcode information printout. The present results can be compared with reports from other countries, where for 89% of the respondents it was important to always receive a paper informational printout of the prescription. In contrast, 32% of patients expected to be able to handle e-prescriptions on a smartphone screen [23]. A small percentage of people, who were mostly college-educated, designated email as the most convenient way to receive a prescription. This form is used by about 5% of physicians. In contrast, paper prescriptions were more likely to be preferred by those with vocational education and were popular among 8.8% of users.

Patients also identified several other prescription options that are practiced by their physicians. In the era of the COVID-19 pandemic, it has become commonplace for many physicians, both general practitioners and specialists, to provide online consultations, which may be considered the reason for the relatively high popularity of providing the prescription access code orally during a phone call. Giving the access code on a piece of paper during the visit is another option. It is important to emphasize that these forms of transferring access to the prescription are allowed only after agreement with the patient and provided that the prescription information contains at least the access key and the name of the medicinal product. In accordance with the provisions of the Act of September the 6th, 2001 Pharmaceutical Law, access data to the prescription should be provided to the patient directly from the person who issued the prescription [29].

In the study group of the present study, the majority (345, 75.7%) of respondents were women. The predominance of this gender may be due to the fact that women are more likely than men to be involved in the purchase of medicinal products, as evidenced by studies seeking to segment the recipients of pharmaceutical services [30], and are more likely than men to be responsible for keeping family members healthy and caring for them [31]. It is also worth noting that women are the dominant group in many published studies conducted on the participation of patients visiting pharmacies [32,33,34].

The predominant group among the respondents were young people with higher education, which may be a limitation when generalizing the results to the entire population. Despite the large number of respondents, the study group does not correspond to the characteristics of the Polish population, because in the study 76.7% of people were women, while in fact there are 51.6% in the population. Moreover, in this study 48% suffered from chronic diseases, whereas in reality this is only about 39%. The age group of >70 years only accounted for 5.9% or responses, whereas in reality Polish people over 65 years old are about 18.72% [35]. Other limitations of this study include the partial online distribution of the questionnaire, which made it difficult to contact the researcher directly in case of any doubts.

## 5. Conclusions

Based on the survey a general conclusion can be drawn that e-prescription is well evaluated by Polish patients. It is a technology that is accepted by society and should be developed. The functionality for redeeming medications on behalf of another person was noted and considered good. Such facilitation should be a priority when designing and implementing e-health tools, as there is a sizable group of digitally or mobile-excluded people who need assistance in dealing with their treatment. It was not possible to accurately determine whether respondents knew what e-prescribing was and what their level of knowledge was. Continued patient education in this area is therefore recommended. 

A large percentage of respondents were positive about being able to obtain a prescription for continued treatment without a personal doctor visit. This type of medical visit became widespread during the COVID-19 pandemic. Based on the data obtained, it is reasonable to conclude that the possibility of such contact with a physician should be maintained in some cases. More than one-fifth of those surveyed still have concerns about the security of their data when using e-prescribing. This indicates the need to raise awareness and intensify educational campaigns on this topic.

The most popular and preferred method of receiving e-prescriptions is via SMS. However, due to the wide variation in responses received, it is necessary to offer different options for obtaining prescriptions, to meet the needs of different populations. In conclusion, hardware, digital applications, and the web should be adapted to each social group, especially the elderly and people with disabilities. Properly designed e-health tools will help avoid potential inequalities in access to healthcare, thus improving public health in Polish society.

## Figures and Tables

**Figure 1 ijerph-18-09769-f001:**
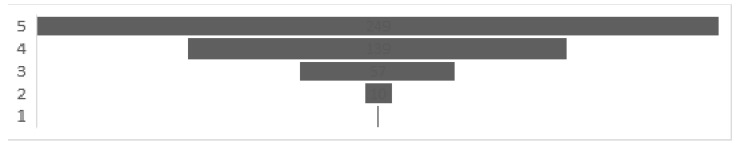
Quantitative distribution of responses subjectively assessing patient satisfaction with e-prescribing.

**Table 1 ijerph-18-09769-t001:** Table presenting the characteristics of the study group.

Sex		Frequency	Percent
	Female	345	75.7
	Male	111	24.3
Overall		456	100
Age			
	<30 years old	191	41.9
	31–50 years old	148	32.5
	51–70 years old	90	19.7
	>70 years old	27	5.9
Overall		456	100
Education			
	Primary	5	1.1
	Vocational	16	3.5
	Secondary	126	27.6
	Higher	309	67.8
Overall		456	100
Place of residence			
	Village	73	16
	Small town (<20,000 residents)	37	8.1
	Medium-sized town (20–100 thousand residents)	59	12.9
	Large city (>100 thousand residents)	287	62.9
Overall		456	100
Chronic diseases			
	Yes	219	48
	No	237	52
Overall		456	100
Do you know what e-prescription is?			
	Yes	448	48
	No	8	1.8
Overall		456	100
Have you filled an e-prescription in the pharmacy in the last six months?			
	Yes	401	89.5
	No	47	10.5
Overall		448	100
Do you use Online Patient Account?			
	Yes	226	49.6
	No	230	50.4
Overall		456	100

**Table 2 ijerph-18-09769-t002:** Table showing the numerical and percentage distribution of responses to the question “How often in the last six months have you dispensed a prescription for another person” by age and in relation to place of residence.

How Often in the Last 6 Months Have You Filled a Prescription for Someone Else															
	Never		Once-Twice		3–4 Times		5–6 Times		>6 Times		Overall		Pearson’s Chi-Squared (*N* = 456)		
	*n*	%	*n*	%	*n*	%	*n*	%	*n*	%	*n*	%	Value	df	*p*
**Age**													32.175	12	0.001
**<30 years old**	89	46.6%	70	36.6%	20	10.5%	5	2.6%	7	3.7%	191	41.9%			
**31–50 years old**	42	28.4%	62	41.9%	25	16.9%	8	5.4%	11	7.4%	148	32.5%			
**51–70 years old**	37	41.1%	26	28.9%	14	15.6%	5	5.6%	8	8.9%	90	19.7%			
**>70 years old**	20	74.1%	3	11.1%	1	3.7%	1	3.7%	2	7.4%	27	5.9%			
**Overall**	188	41.2%	161	35.3%	60	13.2%	19	4.2%	28	6.1%	456	100%			
													Value	df	*p*
**Place of Residence**													27.152	12	0.007
**village**	18	24.7%	27	37.0%	11	15.1%	9	12.3%	8	11.0%	73	16%			
**small town (<20,000 residents)**	14	37.8%	13	35.1%	7	18.9%	2	5.4%	1	2.7%	37	8.1%			
**medium size town (20,000–100,000 residents)**	24	40.7%	21	35.6%	9	15.3%	2	3.4%	3	5.1%	59	12.9%			
**large city (>100,000 residents)**	132	46.0%	100	34.8%	33	11.5%	6	2.1%	16	5.6%	287	63%			
**Overall**	188	41.2%	161	35.3%	60	13.2%	19	4.2%	28	6.1%	456	100%			

**Table 3 ijerph-18-09769-t003:** Numerical summary of responses to the question: when you fill prescriptions for another person, for whom are the drugs prescribed?

When You Fill Prescriptions for Another Person, for Whom Are the Drugs Prescribed?	Number of Respondents
I don’t fill such prescriptions	105
Friends/Acquaintances/Neighbors	18
Siblings	7
Parents	158
Extended family	7
Grandparents	28
Child	83
Partner/Spouse	169

**Table 4 ijerph-18-09769-t004:** Summary of arguments justifying the preference for the form of prescription received.

Arguments Justyying the Preference	Frequency	Percent
Advantages of Electronic Prescriptions		
E-prescriptions are more convenient	124	28.8
No risk of losing or damaging prescriptions	70	16.2
No need for a doctor’s visit	43	10.0
I always have it with me	30	7.0
It is environmentally friendly, greener	30	7.0
Easier access to electronic prescriptions	23	5.3
No prescription illegibility problem	22	5.1
Ease of dispensation	22	5.1
Easier to buy medication for another person	15	3.5
Each drug can be purchased at a different pharmacy	12	2.8
Ability to purchase only some of the prescribed drugs	10	2.3
I have the ability to control my prescribed medications on my OPA	8	1.9
Getting used to having all documents on my phone	5	1.2
Reduced risk of prescription errors	4	0.9
Ability to prescribe a year’s worth of therapy	3	0.7
Easier protection of sensitive data	3	0.7
I have the ability to order prescription medications over the phone and pick them up without a queue	2	0.5
Electronic prescriptions are progress	2	0.5
Reduced risk of prescription forgery	1	0.2
After dispensation, I still have the dosage information printout	1	0.2
E-prescriptions have longer expiration and dispensation times	1	0.2
Overall	431	100
Advantages of paper prescriptions		
Easier to view the list of prescribed drugs	13	35
Having control over prescribed medications	6	16
E-prescriptions are a challenge for older people	5	14
Habit	4	11
Ease of dispensation	3	8
No need to give your PESEL number while buying the medications	3	8
It is easier to buy medicines on behalf of another person (no need to provide the PESEL number)	1	3
Unlike an e-prescription, knowing how long your prescription is valid	1	3
Problem with remembering the pharmacy where the e-prescription was started	1	3
Overall	37	100

**Table 5 ijerph-18-09769-t005:** Percentage distribution of responses evaluating e-prescribing.

Statements about E-Precriptions	Strongly Disagree	Rather Disagree	No Opinion	Rather Agree	Strongly Agree
I think that e-prescriptions increase my autonomy during the therapy	7%	10.5%	36.4%	18.4%	27.60%
With e-prescriptions I have a greater control over the prescribed drugs	11.20%	18.20%	25.70%	20.0%	25.0%
I think that e-prescriptions are eco-friendly	2.2%	3.3%	7.9%	19.5%	67.1%
I think that e-prescriptions enhance the access to drugs	11.6%	16.7%	30%	16.4%	25.2%
With e-prescriptions it is easier to buy drugs for another person	7.9%	9%	21.1%	19.1%	43%
I have concerns over my privacy when using e-prescriptions	35.5%	26.1%	17.8%	10.7%	9.9%

**Table 6 ijerph-18-09769-t006:** Distribution of answers to the question about the most common form of receiving prescriptions.

Form of Receiving Prescriptions	Frequency	Percentage
SMS	254	55.7%
information printout with barcode	87	19.2%
paper-based prescription	40	8.8%
e-mail	23	5%
prescription code given verbally during an online consultation	19	4.2%
prescription code given verbally during an in-person visit	8	1.8%
paper with the prescription code	7	1.5%
mCitizen app	6	1.3%
Online Patient Account	5	1.1%
unclassifiable responses	7	1.5%

## Data Availability

The study did not report any data.

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
