# Peer review of "A Survey of Patients’ Opinions and Preferences on the Use of E-Prescriptions in Poland"

_ijerph, 2021, doi:10.3390/ijerph18189769_

Round 1

Reviewer 1 Report

Report on “A Survey of Patients’ Opinions and Satisfaction on the Use of e-prescriptions in Poland.”

Manuscript Number: ijerph-1365090

Int. J. Environ. Res. Public Health

Summary

The article evaluates satisfaction, features and functionalities of e-prescribing in Poland from a patient perspective.

Assessment

For a successful adoption of e-prescriptions it is important to understand the underlying features and functionalities that are desired by the involved actors. While there have been several studies regarding the perspective of physicians and pharmacists, the assessment of the patient’s perspective is underrepresented in literature. Therefore, the findings of this study enrich the empirical evidence.

The used data set offers a valid approach to address the research questions. However, the description of the applied method and the presentation of the results lack of clarity making it hard for the reader to follow.

In what follows, I give a couple of suggestions that may improve the paper.

Abstract

  • Line 17: What people? What is the population you have drawn from? Please also provide a response rate.
  • Line 19: “because it is” instead of “because is”

Introduction

  • The introduction is very long explaining deeply the implementation of e-prescriptions in Poland over four paragraphs (lines 35-84). This makes it hard for the reader to recognize the focus of the study. From my opinion these details are not relevant for the understanding of your approach, and I would suggest shortening this part of the introduction about the Polish system and to reference the reader for more details to other sources or place it in an extra section (or an appendix).
  • You put very much emphasize in the results and discussion section about filling a prescription for someone else. You should motivate why this aspect is so important in the introduction.

Methods

  • Line 125: Which filter questions were used? And why?

Results

  • Line 147: Please add the response rate.
  • Line 147-159: Where can I find these reported numbers? Please provide a table including sample characteristics.
  • Line 159-181: You consume much space for providing results about filling a prescription for someone else (2 tables and 1 figure). At least tables 1 and 2 could be condensed in one table. Is Figure 1 necessary? These numbers might be shown in a table that consumes much less space.
  • Line 182-191: Where can I find these reported numbers? Please provide a table including these numbers, e.g., in the table with the sample characteristics.
  • Line 196-202: For me, it is not clear how the reported percentages are computed. For instance, in line 199, 13 responses and 25%. In Table 3, there are 23 respondents who mentioned easier access. Are the 13 responses from the group of 25 supporters of traditional paper-based prescriptions (line190)?
  • Table 3: The last-mentioned aspect (Problem with remembering the pharmacy where the e-prescription was started) raises the question how the e-prescription process works. From the detailed description in the introduction, for me it was not clear that the patient has to remember the pharmacy where the e-prescription was started. Need the patient choose a pharmacy prior to receiving the prescription? Could you please clarify this aspect?
  • Line 209: “agreed” not “agree”
  • Figure 2: These numbers could also be condensed by providing just averages in a table. Figure 2 could be replaced in an appendix.
  • Line 218-221: How many persons use OPA? Please provide these numbers in a table.
  • Figure 3: Is this figure necessary? It also consumes much place for the scarce information that it provides. The information could also be included in a table in condensed form.
  • Line 231-238: Where can I find these reported numbers? Please provide a table including these numbers.

Discussion

  • Please provide strengths and weaknesses of your study. For instance, might there be a selection bias in your data. It would be interesting if you compare your numbers about the peoples’ health (see line 154-157) with nationwide statistics when mentioning this potential weakness.
  • Line 257-260: Please provide a reference for this statement.

Author Response

Abstract

Comment 1.

Line 17: What people? What is the population you have drawn from? Please also provide a response rate.

Response 1.

  • The indicated sentence was corrected and the form of selecting the test group was made more precise.
  • Response rate was not calculated as the questionnaire was distributed through many channels, both online and in paper form, in order to obtain a high number of completed forms.

Comment 2.

Line 19: “because it is” instead of “because is”

Response 2.

Corrected as suggested

Introduction

Comment 3.

The introduction is very long explaining deeply the implementation of e-prescriptions in Poland over four paragraphs (lines 35-84). This makes it hard for the reader to recognize the focus of the study. From my opinion these details are not relevant for the understanding of your approach, and I would suggest shortening this part of the introduction about the Polish system and to reference the reader for more details to other sources or place it in an extra section (or an appendix).

Response 3.

Fragments describing in detail the Polish conditions for issuing e-prescriptions, which were of little importance to the main problem of the study, have been removed.

Comment 4.

You put very much emphasize in the results and discussion section about filling a prescription for someone else. You should motivate why this aspect is so important in the introduction.

Response 4.

The functionality enabling the e-prescription to be completed on behalf of another person has been added to the paragraph dealing with the benefits of electronic prescription.

Methods

Comment 5.

Line 125: Which filter questions were used? And why?

Response 5

Two filtering questions were used, which optionally exempted people who had no experience of using an e-prescription or Internet Patient Account from answering some parts of the survey. The paragraph has been supplemented with this clarifying information.

Results

Comment 6.

Line 147: Please add the response rate.

Response 6

Response rate was not calculated as the questionnaire was distributed through many channels, both online and in paper form, in order to obtain a high number of completed forms.

Comment 7

Line 147-159: Where can I find these reported numbers? Please provide a table including sample characteristics.

Response 7.

The group characteristics table has been added

Comment 8

Line 159-181: You consume much space for providing results about filling a prescription for someone else (2 tables and 1 figure). At least tables 1 and 2 could be condensed in one table. Is Figure 1 necessary? These numbers might be shown in a table that consumes much less space

Response 8

Corrected as suggested

Comment 9

Line 182-191: Where can I find these reported numbers? Please provide a table including these numbers, e.g., in the table with the sample characteristics.

Response 9

The above-mentioned data have been entered in Table 1

Comment 10

Line 196-202: For me, it is not clear how the reported percentages are computed. For instance, in line 199, 13 responses and 25%. In Table 3, there are 23 respondents who mentioned easier access. Are the 13 responses from the group of 25 supporters of traditional paper-based prescriptions (line190)?

Response 10

Information has been added explaining how the data is calculated in line: 197-200

Comment 11

Table 3: The last-mentioned aspect (Problem with remembering the pharmacy where the e-prescription was started) raises the question how the e-prescription process works. From the detailed description in the introduction, for me it was not clear that the patient has to remember the pharmacy where the e-prescription was started. Need the patient choose a pharmacy prior to receiving the prescription? Could you please clarify this aspect?

Response 11

The patient also has the option of partially purchasing drugs, but it is important that subsequent packages of the same drug must be continued at the same pharmacy. – this clarification has been added to introduction

Comment 12

Line 209: “agreed” not “agree”

Response 12

Corrected as suggested

Comment 13

Figure 2: These numbers could also be condensed by providing just averages in a table. Figure 2 could be replaced in an appendix.

Response 13

The data in Figure 3 has been reformatted to tabular form

Comment 14

Line 218-221: How many persons use OPA? Please provide these numbers in a table.

Response 14

The above-mentioned data have been entered in Table 1

Comment 15

Figure 3: Is this figure necessary? It also consumes much place for the scarce information that it provides. The information could also be included in a table in condensed form.

Response 15

In order to visualise the distribution of answers to the question, the form of a graph have been maintained. Chart type have changed to save space.

Comment 16

Line 231-238: Where can I find these reported numbers? Please provide a table including these numbers.

Response 16

A table containing the mentioned data has been added

Comment 17

Please provide strengths and weaknesses of your study. For instance, might there be a selection bias in your data. It would be interesting if you compare your numbers about the peoples’ health (see line 154-157) with nationwide statistics when mentioning this potential weakness.

Response 17

The strengths of the study were included in the first paragraph of the discussion part (line: 256-259).

The weaknesses of the study were taken into account in the last paragraph of the discussion section (line 406-409).

Comment 18

Line 257-260: Please provide a reference for this statement.

Response 18

Publication number 18 is the source of these data.

Discussion

  • Line 257-260: Please provide a reference for this statement.

Reviewer 2 Report

The authors did a good job in describing the feelings and opinions of Polish citizens concerning the Polish e-prescription. On the other hand, in the title, the main focus is on the "opinion" and on the "satisfaction" of Polish citizens. How much I liked reading the work, the more I found that the satisfaction part of the title might not be completely fulfilled or only in a very small part. Therefore, I would like to suggest the authors changing the title to "A Survey of Patients’ Opinions and Preferences on the Use of e-prescriptions in Poland."

1.Introduction
==============

(1.A) In the background material, the authors mainly focused on one review (reference 1), besides a paper of India and Nigeria, to say where the e-prescription serves as a basic tool of e-health. Though in our opinion, these are not the only countries frequently using the e-prescription (especially not in Europe). I lacked some references to Finland, Croatia, Belgium, ... Now it seems as if Spain, Sweden and Denmark are the only European countries using the e-prescription.

A starting point as a reference might be:
European Commission. Electronic cross-border health services. https://ec.europa.eu/health/ehealth/electronic_crossborder_healthservices_en

(1.B) Also in our opinion, the Polish system is insufficiently prescribed in detail. Moreover, in the discussion, the authors state that the Polish system in many ways resembles the Belgian system. I haven't really seen references to this system and what the resemblances are. Please add this in your work.

After a quick look-up, possible references that include a description of the Belgian system include:
- Van Laere S, Cornu P, Buyl R. A cross-sectional study of the Belgian community pharmacist's satisfaction with the implementation of the electronic prescription. Int J Med Inform. 2020 Mar;135:104069. doi: 10.1016/j.ijmedinf.2019.104069. Epub 2019 Dec 28. PMID: 31915117.
- Van Laere S, Cornu P, Dreesen E, Lenie J, Buyl R. Why do Belgian Community Pharmacists Still Treat Electronic Prescriptions as Paper-Based? J Med Syst. 2019 Oct 23;43(11):327. doi: 10.1007/s10916-019-1456-5. PMID: 31646400.

A resemblance between the two countries is for example the large diversity of pharmacy and medical-office software.

(1.C) Is it possible to re-write the part "This is very important because, in the first period of operation of e-prescription, a common problem was the lack of awareness about the need to show ID at the pharmacy when buying drugs so patients, especially the elderly, were confused and it impeded them buying drugs in the pharmacy" (lines 79-82)? Do you mean that since multiple e-prescriptions are linked to the ID of the patient, the patient might get confused since there might be multiple prescriptions that can be retrieved by the pharmacy. Please clarify.

(1.D) Please also add something in text about the possibility to get medication for another person here. The reason I would already write something of this here is that later on it is used in your results and discussion section.

2. Materials and methods
========================

(2.A) I did not completely have the feeling that I know what was asked in the 17-question survey. Or the authors should think providing the questions more in detail, or the authors might opt providing a translated version of it in appendices.

(2.B) Methodology described that a content validity ratio was used and that items were removed when they were less than 0.9. But the authors do not state how many experts were involved in the process of reviewing the self-assembled questionnaire. Optionally, if authors find this relevant, might indicate what questions were removed eventually.

3. Results
==========

(3.A) The main problem with the result section that I faced, was not providing a kind of structure in it, which made it harder to read. The results start to discuss with a general description of the sample, describe about who fills prescriptions and how often, then talk about knowledge and the the use of the e-prescription, than about benefits and drawbacks perceived, etc. I would prefer seeing this in different subparagraphs with an appropriate subtitle, since now I had the feeling that we jumped from one item in the other. 

(3.B) How does the author deal with small cell risk? For example in Table 1 the table is sometimes populated with 1 or 2 people. Wouldn't it be useful to regroup ages 51-70 y.o. together with >70 y.o. since the group of 70 y.o. only provides a subsample of 5.9% or do you think this regrouping will not influence your results significantly?

(3.C) Both table 1 and table 2 provide a row overall, where all columns result in a 100%. Please adjust such that each column percentage represents the number of people divided by the total. Pleas adjust.

(3.D) Two times I found a sentence like "statistically significant relationships were found between variable X (p = p-value_to_report)". The fact that the authors write the word "between" induces that two variables are in fact expected. To be more precise, you can find examples on lines 159 and 185.

For example line 159 can be corrected as follows: "Statistically significant relationships were found between the age and the number of times one dispensed a prescription for someone else (p = 0.001). The relation indicates that ..."

(3.E) When bivariable analyses are reported for a 2x2 table in terms of one group has more of a property compared to another, e.g. see line 186 and 187, the authors might opt using odds ratios with confidence interval tho express how much more/less. Now it remains rather vague.

(3.F) In table 3 some rows are colored in gray. I think the authors want to stress the most relevant items by this. It might be relevant ordering that table (both the upper and the lower part) in terms of prevalence with the highest prevalent arguments each time on top.

Moreover as a detail, the authors only address 3 positive and  negative arguments in text, which does not justify coloring 5 positive arguments and 4 negative arguments in table 3.

4. Discussion
=============

(4.A) Are there already studies that provide proof for the following statement: "This correlation may indicate the need for younger people to support the elderly who may be experiencing e-exclusion, also known as a social exclusion in the information society"?
I find the topic of e-exclusion highly relevant for your argumentation in the discussion. Please provide a reference if there is one/multiple ones. 

(4.B) Lines 283-284: The survey noted only 5.6% supporters of the traditional paper prescription.
This percentage should be corrected to 8.8%.

(4.C) Lines 297-298: "It is worth emphasizing that the e-prescription system in Belgium is most similar to that in Poland."
Please move to the introduction. Possibly when talking about similarities with the Belgian system. 

(4.D) In the limitation part, the authors address the generalisability, but do not really state why. Possible explanation easily found on the web are that: 
- In this study 75.7% are women, whereas this is only 51.6% in reality.
- In this study 48% suffer from chronic diseases, whereas in reality this is only about 39%.
- In this study the age group of >70 years was only populated for 5.9%, whereas in reality Polish people over 65 years old are about 18.72%.

5. Conclusion
=============

(5.A) Reference to the COVID-19 pandemic may already be raised in the introduction. E-prescribing and e-consultation only got more important during the pandemic.

Some minor remarks:
===================

- Line 53: "e-healthis" should be replaced by "e-health is"
- What is PESEL? (first occurrence line 76)
- Difficult to read, lines 77-79 - please re-write
- On line 218 the wording "autonomy" is used, whereas in Figure 2 the wording "independence" is used. Please be consistent.

Author Response

Comment 1.

The authors did a good job in describing the feelings and opinions of Polish citizens concerning the Polish e-prescription. On the other hand, in the title, the main focus is on the "opinion" and on the "satisfaction" of Polish citizens. How much I liked reading the work, the more I found that the satisfaction part of the title might not be completely fulfilled or only in a very small part. Therefore, I would like to suggest the authors changing the title to "A Survey of Patients’ Opinions and Preferences on the Use of e-prescriptions in Poland."

Response 1.

Changed as suggested

1.Introduction
==============

Comment (1.A)

In the background material, the authors mainly focused on one review (reference 1), besides a paper of India and Nigeria, to say where the e-prescription serves as a basic tool of e-health. Though in our opinion, these are not the only countries frequently using the e-prescription (especially not in Europe). I lacked some references to Finland, Croatia, Belgium, ... Now it seems as if Spain, Sweden and Denmark are the only European countries using the e-prescription.

A starting point as a reference might be:
European Commission. Electronic cross-border health services.
https://ec.europa.eu/health/ehealth/electronic_crossborder_healthservices_en

Response (1.A)

Additional data has been added as suggested in the first paragraph (lines: 34-41)

Comment (1.B)

Also in our opinion, the Polish system is insufficiently prescribed in detail. Moreover, in the discussion, the authors state that the Polish system in many ways resembles the Belgian system. I haven't really seen references to this system and what the resemblances are. Please add this in your work.

Response (1.B)

Additional information describing the e-prescription system in Poland is provided on lines 71-76

Comment 2.

After a quick look-up, possible references that include a description of the Belgian system include:
- Van Laere S, Cornu P, Buyl R. A cross-sectional study of the Belgian community pharmacist's satisfaction with the implementation of the electronic prescription. Int J Med Inform. 2020 Mar;135:104069. doi: 10.1016/j.ijmedinf.2019.104069. Epub 2019 Dec 28. PMID: 31915117.
- Van Laere S, Cornu P, Dreesen E, Lenie J, Buyl R. Why do Belgian Community Pharmacists Still Treat Electronic Prescriptions as Paper-Based? J Med Syst. 2019 Oct 23;43(11):327. doi: 10.1007/s10916-019-1456-5. PMID: 31646400.

A resemblance between the two countries is for example the large diversity of pharmacy and medical-office software.

Response 2.

the fragment about the similarity between the e-prescription system in Belgium and the Polish one was changed and clarified (line 333-335)

Comment (1.C)

Is it possible to re-write the part "This is very important because, in the first period of operation of e-prescription, a common problem was the lack of awareness about the need to show ID at the pharmacy when buying drugs so patients, especially the elderly, were confused and it impeded them buying drugs in the pharmacy" (lines 79-82)? Do you mean that since multiple e-prescriptions are linked to the ID of the patient, the patient might get confused since there might be multiple prescriptions that can be retrieved by the pharmacy. Please clarify.

Response (1.C)

An explanatory sentence has been added (line: 92-93)

Comment (1.D)

Please also add something in text about the possibility to get medication for another person here. The reason I would already write something of this here is that later on it is used in your results and discussion section.

Response (1.D)

An excerpt regarding the possibility of dispensing a prescription on behalf of another person has been added on lines 54-57

  1. Materials and methods
    ========================

Comment (2.A)

I did not completely have the feeling that I know what was asked in the 17-question survey. Or the authors should think providing the questions more in detail, or the authors might opt providing a translated version of it in appendices.

Response (2.A)

A translated version of the questionnaire was added as an attachment.

Comment (2.B)

Methodology described that a content validity ratio was used and that items were removed when they were less than 0.9. But the authors do not state how many experts were involved in the process of reviewing the self-assembled questionnaire. Optionally, if authors find this relevant, might indicate what questions were removed eventually.

Response (2.B)

The tool was validated by five independent experts from the Medical University of Gdansk with experience in social research. Content Validity Ratio (CVR) was determined for each question separately. - as written on lines 138 -140.

  1. Results
    ==========

Comment (3.A)

The main problem with the result section that I faced, was not providing a kind of structure in it, which made it harder to read. The results start to discuss with a general description of the sample, describe about who fills prescriptions and how often, then talk about knowledge and the the use of the e-prescription, than about benefits and drawbacks perceived, etc. I would prefer seeing this in different subparagraphs with an appropriate subtitle, since now I had the feeling that we jumped from one item in the other. 

Response (3.A)

Corresponding subheadings in the results section have been added. Extremely high or low values ​​in cells illustrate differences in responses in the respective groups.

Comment (3.B)

How does the author deal with small cell risk? For example in Table 1 the table is sometimes populated with 1 or 2 people. Wouldn't it be useful to regroup ages 51-70 y.o. together with >70 y.o. since the group of 70 y.o. only provides a subsample of 5.9% or do you think this regrouping will not influence your results significantly?

Response (3.B)

These tables were intended to show the differences in responses depending on age or place of residence. Extremely high or low values ​​in cells illustrate differences in responses in the respective groups. Low cell values ​​are not used in further calculations so should not negatively affect the rest of the results.

Comment (3.C)

Both table 1 and table 2 provide a row overall, where all columns result in a 100%. Please adjust such that each column percentage represents the number of people divided by the total. Pleas adjust.

Response (3.C)

The tables have been modified and combined into one. In each row, the sum of the percentages is 100%, and the sum of the responses equals the number in the last column.

Comment (3.D)

Two times I found a sentence like "statistically significant relationships were found between variable X (p = p-value_to_report)". The fact that the authors write the word "between" induces that two variables are in fact expected. To be more precise, you can find examples on lines 159 and 185.

For example line 159 can be corrected as follows: "Statistically significant relationships were found between the age and the number of times one dispensed a prescription for someone else (p = 0.001). The relation indicates that ..."

Response (3.D)

Corrected as suggested (lines 182 and 206)

Comment (3.E)

When bivariable analyses are reported for a 2x2 table in terms of one group has more of a property compared to another, e.g. see line 186 and 187, the authors might opt using odds ratios with confidence interval tho express how much more/less. Now it remains rather vague.

Response (3.E)

The paragraph was supplemented with Odd ratio and confidence level (line 209)

Comment (3.F)

In table 3 some rows are colored in gray. I think the authors want to stress the most relevant items by this. It might be relevant ordering that table (both the upper and the lower part) in terms of prevalence with the highest prevalent arguments each time on top.

Moreover as a detail, the authors only address 3 positive and  negative arguments in text, which does not justify coloring 5 positive arguments and 4 negative arguments in table 3.

Response (3.F)

The table has been revised according to the comments

  1. Discussion
    =============

Comment (4.A)

Are there already studies that provide proof for the following statement: "This correlation may indicate the need for younger people to support the elderly who may be experiencing e-exclusion, also known as a social exclusion in the information society"?
I find the topic of e-exclusion highly relevant for your argumentation in the discussion. Please provide a reference if there is one/multiple ones. 

Response (4.A)

Reference 19 deals with the topic of e-exclusion and is the source for these statements.

Comment (4.B)

 Lines 283-284: The survey noted only 5.6% supporters of the traditional paper prescription.
This percentage should be corrected to 8.8%.

Response (4.B)

According to the results presented in paragraphs 212-215, the number of supporters of a paper prescription is 25 people, which is 5.6% of all respondents declaring knowledge of electronic prescriptions, who can therefore distinguish between a paper prescription and an electronic prescription.

Comment (4.C)

Lines 297-298: "It is worth emphasizing that the e-prescription system in Belgium is most similar to that in Poland." Please move to the introduction. Possibly when talking about similarities with the Belgian system. 

Response (4.C)

The fragment concerning the similarity between the Belgian and Polish e-prescription system has been clarified. Marked with the appropriate reference

Comment (4.D)

In the limitation part, the authors address the generalisability, but do not really state why. Possible explanation easily found on the web are that: 
- In this study 75.7% are women, whereas this is only 51.6% in reality.
- In this study 48% suffer from chronic diseases, whereas in reality this is only about 39%.
- In this study the age group of >70 years was only populated for 5.9%, whereas in reality Polish people over 65 years old are about 18.72%.

Response (4.D)

The limitation part has been revised according to the comments.

  1. Conclusion
    =============

Comment (5.A)

Reference to the COVID-19 pandemic may already be raised in the introduction. E-prescribing and e-consultation only got more important during the pandemic.

Response (4.D)

The fragment concerning the COVID-19 pandemic has been added to introduction part (line: 54).

Some minor remarks:
===================

  1. Line 53: "e-healthis" should be replaced by "e-health is"

- corrected

  1. What is PESEL? (first occurrence line 76)

-The explanation has been added (line 86)

  1. Difficult to read, lines 77-79 - please re-write

-corrected

  1. On line 218 the wording "autonomy" is used, whereas in Figure 2 the wording "independence" is used. Please be consistent.

-corrected

Round 2

Reviewer 1 Report

The authors addressed all my previously raised concerns.

Author Response

Thank you for your positive review and all valuable comments.

Reviewer 2 Report

The authors did a magnificent job in adjusting the manuscript based on the comments made. However, some comments were only partly answered or were handled a bit sloppy in the way of writing/editing (e.g. new typos were introduced, comments were misunderstood, ...). Moreover, the line numbers referred  to were not always correct in providing there answer, which made it sometimes hard for reviewing.

Comment 2 and 4C
=============

Again a reference to the Belgian system was made to the work of Suykerbuyk et al. (2018) It is not a bad reference, but as suggested in the previous round, I deem the references to the work of Van Laere et al. more appropriate when talking about the technical aspects. A more technical reader of your future article (if accepted) might benefit from this reference in that way that Suykerbuyk only gave a brief description of the system in a paper which not really argues about the technicalities.

Moreover, I would also like to ask if there is yet no such documentation about the Polish system? Because if you talk about a resemblance and you only refer to the Belgian system, I would expect to have a Polish reference as well. It looks a bit odd now. But, if no scientific literature exists yet about the Polish system, there is no further problem.

Comment 2A
=========

Thanks for providing the questionnaire in Appendix. However, only question 17 did provide the Polish translation. Or you do it for all questions, or you leave it at question 17. Consistency is important to me, especially publishing in a renowned journal.

Comment 2B
=========

line 142: Appendix 1 refers to the full questionnaire and not to the items left out by CVR. Please move the reference to appendix 1 to line 133.

Comment 3B
=========

By changing table 2, we saw a typo entered:

"2,.6%" should be corrected to "2.6%"

Also in table 1, we sometimes observed sometimes using a decimal "," instead of a decimal "."

Please check other tables as well.

Comment 3C
=========

I believe the table significantly improved, however, the reader still needs to count himself how many respondents answered "never", "once-twice", ... I believe my previous comment was misunderstood partly.

I would like to have both the row and the column numbers, so both an overall row/column for both directions. Once this is provided, I would like you to provide the percentages for these numbers towards the overall frequency. 

A good starting point was your original submitted version (there labelled table 1), where the overall column of the first part contained "188 (41.2%)", "161 (35.3%)", "60 (13.2%)", "19 (4.2%)" and "28 (6.1%)". The only problem over there I had, was the row "Overall" that each time resulted in 100%. I would have expected the following numbers "191 (41.9%)", "148 (32.5%)", "90 (19.7%)" and "27 (5.9%)".

The same technique of providing relative frequencies might be applied to the second part as well, since it is makes more sense. Please provide likewise for the whole of the new table 2. One row needs to be added at the bottom and the percentage column underneath the overall column might change to the percentages provided in the original submitted manuscript.

Comment 3E
=========

new typo arose in line 214: "Odd ratio" should change to "Odds ratio"

Comment 4D
=========

new typo arose in line 450: "Moreover, nn this study 48% suffer ..." should change to "Moreover, in this study 48% suffer ..."

Author Response

The authors did a magnificent job in adjusting the manuscript based on the comments made. However, some comments were only partly answered or were handled a bit sloppy in the way of writing/editing (e.g. new typos were introduced, comments were misunderstood, ...). Moreover, the line numbers referred  to were not always correct in providing there answer, which made it sometimes hard for reviewing.

Comment 2 and 4C
=============

Again a reference to the Belgian system was made to the work of Suykerbuyk et al. (2018) It is not a bad reference, but as suggested in the previous round, I deem the references to the work of Van Laere et al. more appropriate when talking about the technical aspects. A more technical reader of your future article (if accepted) might benefit from this reference in that way that Suykerbuyk only gave a brief description of the system in a paper which not really argues about the technicalities.

  • The proposed source was added to the manuscript as reference number 2

Moreover, I would also like to ask if there is yet no such documentation about the Polish system? Because if you talk about a resemblance and you only refer to the Belgian system, I would expect to have a Polish reference as well. It looks a bit odd now. But, if no scientific literature exists yet about the Polish system, there is no further problem.

  • Until then, no scientific publication comprehensively presenting the technical aspects of the e-prescription has been published. The way the e-prescription works in Poland is described in paragraphs 79-96, taking into account the most important facts relevant to the topic of the work.
  • It is worth to emphasize that the topic of the work is not a detailed analysis of the technical aspects of the tool, but the opinion and perception of patients. From the patient's point of view, the detailed operation of the e-prescription system is not important, but its practical application. After an previous review, I specified in lines 342-345 that the similarity of the Polish and Belgian e-prescription concerns only the way the patient downloads it (using an information printout with a barcode).

 Comment 2A
=========

Thanks for providing the questionnaire in Appendix. However, only question 17 did provide the Polish translation. Or you do it for all questions, or you leave it at question 17. Consistency is important to me, especially publishing in a renowned journal.

  • The English translation was corrected and completed 

Comment 2B
=========

line 142: Appendix 1 refers to the full questionnaire and not to the items left out by CVR. Please move the reference to appendix 1 to line 133.

  • Corrected as suggested

 Comment 3B
=========

By changing table 2, we saw a typo entered:

"2,.6%" should be corrected to "2.6%"

  • corrected

Also in table 1, we sometimes observed sometimes using a decimal "," instead of a decimal "."

  • corrected

Please check other tables as well.

  • checked

Comment 3C
=========

I believe the table significantly improved, however, the reader still needs to count himself how many respondents answered "never", "once-twice", ... I believe my previous comment was misunderstood partly.

I would like to have both the row and the column numbers, so both an overall row/column for both directions. Once this is provided, I would like you to provide the percentages for these numbers towards the overall frequency. 

A good starting point was your original submitted version (there labelled table 1), where the overall column of the first part contained "188 (41.2%)", "161 (35.3%)", "60 (13.2%)", "19 (4.2%)" and "28 (6.1%)". The only problem over there I had, was the row "Overall" that each time resulted in 100%. I would have expected the following numbers "191 (41.9%)", "148 (32.5%)", "90 (19.7%)" and "27 (5.9%)".

The same technique of providing relative frequencies might be applied to the second part as well, since it is makes more sense. Please provide likewise for the whole of the new table 2. One row needs to be added at the bottom and the percentage column underneath the overall column might change to the percentages provided in the original submitted manuscript.

  • corrected as suggested

Comment 3E
=========

new typo arose in line 214: "Odd ratio" should change to "Odds ratio"

  • corrected

Comment 4D
=========

new typo arose in line 450: "Moreover, nn this study 48% suffer ..." should change to "Moreover, in this study 48% suffer ..."

  • corrected
